# Genomic locus proteomic screening identifies the NF-κB signaling pathway components NFκB1 and IKBKG as transcriptional regulators of *Ripk3* in endothelial cells

Siqi Gao[1,2¤a], Matthew Menendez[1], Katarzyna Kurylowicz[1¤b], Courtney T. Griffin[1,2] *

**1** Cardiovascular Biology Research Program, Oklahoma Medical Research Foundation, Oklahoma City, Oklahoma, United States of America, **2** Department of Cell Biology, University of Oklahoma Health Sciences Center, Oklahoma City, Oklahoma, United States of America

¤a Current address: Department of Medicine and Cardiovascular Institute, University of Pennsylvania, Philadelphia, Pennsylvania, United States of America
¤b Current address: The Ben May Department for Cancer Research, University of Chicago, Chicago, Illinois, United States of America
* Courtney-Griffin@omrf.org

**Data Availability Statement:** The minimal data set is available within the paper and its Supporting Information files. The GLoPro data generated

## Abstract

The receptor-interacting protein kinase 3 (RIPK3) is a multi-functional protein best known for facilitating cellular necroptosis and inflammation. Recent evidence from our lab indicates that RIPK3 expression must be tightly regulated in endothelial cells to promote angiogenesis, to maintain vascular integrity during embryogenesis, and to provide protection from postnatal atherosclerosis. RIPK3 activity and stability are regulated by post-translational modifications and caspase-dependent cleavage. However, less is known about the transcriptional regulation of *Ripk3*. Here we utilized an unbiased CRISPR-based technology called genomic locus proteomics (GLoPro) to screen transcription factors and coregulatory proteins associated with the *Ripk3* locus in a murine endothelial cell line. We found that 41 nuclear proteins are specifically enriched at the *Ripk3* locus, including the Nuclear Factor kappa-light-chain-enhancer of activated B cells (NF-κB) signaling pathway components NFκB1 and IKBKG. We further verified that NFκB1 and IKBKG directly bind the *Ripk3* promoter and prevent TNFα-induced *Ripk3* transcription in cultured human primary endothelial cells. Moreover, NFκB1 prevents RIPK3-mediated death of primary endothelial cells. These data provide new insights into NF-κB signaling and *Ripk3* transcriptional regulation in endothelial cells.

## Introduction

RIPK3 is a member of the receptor-interacting protein kinase (RIP) family of Ser/Thr protein kinases [1]. This broadly expressed protein is a critical executioner kinase in the necroptosis programmed cell death pathway [2]. RIPK3 also promotes inflammation through both kinase-dependent and -independent mechanisms [3–5]. Notably, overexpression of RIPK3 in various

during this study have been deposited to the ProteomeXchange Consortium via the PRIDE partner repository at: http://www.ebi.ac.uk/pride. The accession number is: PXD025675.

**Funding:** This work was supported by grants from the National Institutes of Health (https://www.nih.gov) awarded to C.T.G. (R35HL44605) and from the American Heart Association (https://www.heart.org) awarded to S.G. (#19PRE34380708). The funders had no role in study design, data collection and analysis, decision to publish, or preparation of the manuscript.

**Competing interests:** The authors have declared that no competing interests exist.

cell types can promote cell death and inflammatory cytokine production [6–9]. We recently reported that the chromatin-remodeling enzyme CHD4 suppresses *Ripk3* expression in hypoxic murine embryonic endothelial cells (ECs) and that excessive RIPK3 expression in *Chd4*-deficient ECs contributes to lethal vascular rupture [10]. Interestingly, this RIPK3-mediated vascular rupture is not obviously linked to EC necroptosis or inflammation, indicating that aberrant RIPK3 expression can contribute to other damaging processes in ECs. Conversely, genetic deletion of endothelial *Ripk3* impairs developmental angiogenesis [11] and exacerbates vascular lesion formation in a murine atherosclerosis model [12]. Therefore, endothelial RIPK3 expression must be tightly regulated to maintain vascular integrity, function, and homeostasis at different stages of life.

The activity and stability of RIPK3 are controlled by caspase-dependent cleavage and by various post-translational modifications, including phosphorylation, ubiquitination, and glycosylation [13–16]. However, little is known about the transcriptional regulation of *Ripk3* in any cell type, other than its repression by CHD4 and by hypoxia-inducible factor 1 (HIF-1) in hypoxic ECs [10], its repression by CHD4 in muscle stem cells [17], and its methylation-dependent repression and promotion by the transcription factor SP1 in tumor cells [18,19]. Therefore, we sought to identify transcriptional regulators of *Ripk3* in this study, and we chose to perform our screen in ECs because of our interest in the detrimental impact of misregulated RIPK3 expression on embryonic and postnatal vasculature.

Chromatin immunoprecipitation (ChIP) is an invaluable technique for identifying genetic targets of known transcriptional regulatory proteins. However, ChIP is not useful for identifying unknown transcriptional regulatory proteins that interact with a specific genomic locus. By contrast, genomic locus proteomics (GLoPro) is a newly reported and unbiased technique, which combines CRISPR-based genome targeting, proximity biotin-labeling, and quantitative proteomics to identify multiple proteins associated with a predefined genomic locus in living cells [20]. Here we utilized GLoPro technology to screen for transcription factors and coregulatory proteins associated with the *Ripk3* locus in a cultured murine EC line, and we validated two of our findings—members of the NF-κB signaling pathway—in primary human ECs.

The NF-κB signaling pathway balances fundamental cellular survival and pro-inflammatory responses in a variety of cell types and is comprised of multiple transcription factor subunits and cytoplasmic proteins that regulate their nuclear translocation [21]. Activation of canonical NF-κB signaling can be triggered by inflammatory stimuli such as lipopolysaccharides (LPS), tumor necrosis factor α (TNFα), or interleukin-1β [22]. In ECs, a major consequence of NF-κB signaling activation is the robust expression of adhesion molecules that mediate the capture and extravasation of inflammatory cells from the circulation and into underlying tissues [23]. Importantly, NF-κB signaling also promotes EC survival under inflammatory conditions, which counterbalances the pro-permeability effects of cytokines and prevents catastrophic hemorrhage [24]. Our following discovery that members of the NF-κB signaling pathway transcriptionally regulate *Ripk3* expression in stimulated ECs provides new insights into the relationship between NF-κB and RIPK3 and their known roles at the intersection between inflammatory and cell death pathways.

## Materials and methods

### Cell culture

The MS1 adult murine pancreatic endothelial cell line (ATCC) was maintained at 37°C and 5% $CO_2$ in Dulbecco's Modified Eagle Medium (DMEM) containing 5% fetal bovine serum. Authentication of the MS1 line was performed by ATCC, and the sex of MS1 cells is unknown, according to the ATCC manual. MS1 cells were electroporated with the iCaspex plasmid using

a Gene Pulser II system (BioRad; 250 V, 500 uF) in serum-free OptiMEM (Invitrogen). After iCaspex electroporation, clonal selection was performed for 2 weeks under 2μg/mL puromycin. Single colonies were picked and tested for doxycycline inducibility by GFP expression. HEK293T cells were used for lentivirus production and were cultured at 37˚C and 5% $CO_2$ in DMEM containing 10% fetal bovine serum. Plasmids encoding gRNAs, gag, pol, tat, or VSVG were transfected into HEK293T cells with polyethylenimine. Two days later, cell culture supernatants containing lentivirus were collected and filtered through a 0.45 μm filter. sgRNA lentiviruses were transduced into MS1-Caspex cells in the presence of 10 μg/mL polybrene and were selected for stable incorporation after growth in 400 μg/mL hygromycin for 2 weeks. HUVECs (ATCC) were cultured in EBM-2 media supplemented with EGM and 10% fetal bovine serum according to the manufacturer's recommendation (Lonza). All HUVECs used in this study were analyzed at passage 4–6. For siRNA-mediated gene knockdown, HUVECs were transfected with NFκB1 siRNA oligos (Thermo Fisher Scientific; #s9505), IKBKG siRNA oligos (Thermo Fisher Scientific; #s16186), RIPK3 siRNA oligos (Thermo Fisher Scientific; #136148 or #110923), or non-specific controls siRNA oligos (Thermo Fisher Scientific; #4390844) using Lipofectamine RNAiMAX reagent (Thermo Fisher Scientific) according to the manufacturer's protocol.

## Plasmid construction

sgRNA oligonucleotides targeting the promoter of murine *Ripk3* were designed using the Broad Institute GPP Web Portal and were cloned into the pLenti SpBsmBI sgRNA Hygro plasmid. Cloning primers are listed in S6 Table. Plasmids are listed in S9 Table. All modified plasmids were subjected to DNA sequencing for verification. All constructs were prepared using Maxi Prep kits (Qiagen).

## MS1 RIPK3 knockout (KO) cell line generation

RIPK3-KO MS1 cells were generated with CRISPR/Cas9 technology. Briefly, sgRNAs targeting the coding region of *Ripk3* (S6 Table) were designed using the CHOPCHOP website and cloned into the CRISPR plasmid pSpCas9 (BB)-2A-GFP via the BbsI site. GFP-positive cells were sorted into a 96-well plate at a density of 1 cell per well with a FacsAria IIIu cell sorter (BD Biosciences). After the colonies were expanded, RIPK3 KO was validated by immunoblotting.

## Immunoblots

Cells were lysed in RIPA buffer with Protease Inhibitor Cocktail (Thermo Fisher Scientific). Protein concentration was determined using the Pierce BCA Protein Assay Kit. Protein was electrophoresed on a 10% SDS-PAGE gel and then transferred to a PVDF membrane that was blocked in 5% nonfat dry milk-TBST for 1 hr. Primary antibodies (diluted in 5% milk-TBST) were incubated at 4˚C overnight with gentle agitation, and membranes were then washed three times (15 min each) in TBST. HRP-conjugated secondary antibodies (diluted in 5% milk-TBST) were applied at room temperature for 1 hr with gentle agitation, and membranes were then washed five times (15 min each) in TBST. Secondary antibodies were detected using ECL western blotting detection reagents. All antibodies are listed in S5 Table.

## Cell death analysis

Floating and attached HUVECs were harvested and centrifuged at 350 g for 5 min. Cells were then incubated with propidium iodide (BD Pharmingen) on ice for 1 min in the dark. Flow cytometric analysis was performed with a BD FACSCelesta flow cytometer.

## Enrichment of biotinylated proteins for LC-MS/MS

Four T175 flasks of each sgRNA-dCas9-APEX2 EC line (~1X10$^8$ cells per line) and four flasks of the non-targeting gRNA (negative control) were grown for proteomic experiments. 500 ng/mL doxycycline was added to the cell culture media for 24 hours to induce the expression of dCas9-APEX2. Subsequently, 500 μm biotin tyramide phenol (APExBIO) in DMSO was added directly to the cell culture media. After 30 minutes, 1mM hydrogen peroxide was added to the media to induce biotinylation. After 60 seconds of very gentle swirling, the media was discarded, and the cells were washed three times with ice cold PBS containing 100 mM sodium azide and 100 mM sodium ascorbate. Cells were scraped and pelleted in 15 ml Falcon tubes with ice cold PBS. Biotinylated whole cell pellets were lysed with RIPA (50 mM TRIS pH 8.0, 1% Triton X-100, 1% sodium deoxycholate, 0.1% sodium dodecyl sulfate, and 150 mM NaCl) containing a protease inhibitor cocktail and sonicated to shear genomic DNA using a Misonix S-4000 sonicator (five 10 s pulses at an amplitude of 50). Whole cell lysates were clarified by centrifugation at 12,000g for 30 minutes at 4˚C. Lysates of equal protein amounts from each gRNA line were incubated with 250 μL of a streptavidin magnetic bead slurry for 2 hours at 4˚C and subsequently washed three times with cold lysis buffer. Bead-bound proteins were digested and subjected to subsequent LC-MS/MS analysis. Each protein sample was analyzed using three replicate LC-MS/MS analyses on a Fusion model quadrupole-Orbitrap mass spectrometer equipped with an Easy-nLC 1200 UPLC system and a Nanospray Flex ion source (Thermo). Proteins were identified using the MaxQuant v1.6.2.10 to search the MS data against the Uniprot Mouse database. Protein hits were analyzed in Perseus 1.6.12.0 (Max Planck Institute of Biochemistry). The mass spectrometry proteomics data have been deposited to the ProteomeXchange Consortium via the PRIDE partner repository with the dataset identified PXD025675.

## Gene Ontology analyses

Gene Ontology (GO) analyses for enriched 'biological process' terms of high-stringency filtered protein candidates were performed using the web tool DAVID Bioinformatics Database v6.8 (https://david.ncifcrf.gov/tools.jsp) for the 'GOTERM_BP_DIRECT' category. The complete set of all murine genes was used as a background. The DAVID outputs are shown in Fig 3. For KEGG pathway analysis, high-stringency filtered protein candidates were subjected to the web tool g:Profiler and included a false discovery rate of $P<0.05$ after Benjamini-Hochberg correction for multiple comparisons.

## Quantitative Real-Time PCR (qRT-PCR)

Total RNA was isolated from MS1 cells or HUVECs using the RNeasy Mini kit (Qiagen). cDNA was prepared using the iSCRIPT™ Reverse Transcriptase Kit (Bio-Rad), and qRT-PCR was performed using 2X SYBR green qPCR master mix (Applied Biosystems) and the CFX96 Real-Time System (Bio-Rad) with gene-specific primers (S7 Table). The relative fold change in transcription was determined using the comparative C$_T$ method and three housekeeping genes as internal controls: *Gapdh*, *Rn18s*, and *β-actin (Actb)*. Statistical differences were calculated in GraphPad Prism 8. Statistical analyses are detailed in the figure legends.

## Immunocytochemistry

Approximately 2 X 10$^4$ HUVECs were seeded per well of a 24 well plate and cultured on coverslips. After 24 hr, media containing vehicle or TNFα (100 ng/mL) was added to each well and cultured for an additional 24 hr. Fixation was performed by adding a 1% PFA solution in 1X

PBS to monolayers and incubating samples for 5 min at 37˚C. Fixation was quenched by washing samples with a 0.5 M glycine solution (in 1X PBS) two times, 5 min each, at room temperature. Samples were permeabilized by incubating the monolayer with 0.1% Triton X-100 in 1X PBS for 10 min at room temperature. Samples were incubated with 3% BSA with 0.02% NaN$_3$ in 1X PBS (3% BSA) for 1 hr at room temperature to block. Anti-IKBKG Ab was resuspended (1:250) in 3% BSA solution and incubated at 4˚C for 16 hr. Samples were washed 3 times in cold 1X PBS and incubated with anti-rabbit Cy3 (1:500) and Hoechst 10 mg/mL (1:500) in 3% BSA for 1 hr at room temperature. Samples were washed 3 times with cold 1X PBS and mounted onto glass slides using Prolong Gold mounting media. Images were collected using a Nikon C2 Confocal microscope and analyzed using NIS-Elements software.

## Chromatin immunoprecipitation assay

ChIP experiments were performed using the Active Motif ChIP-IT® Express Kit per manufacturer's instructions. Briefly, 2.5 X $10^6$ HUVECs were seeded in a 10 cm dish. Media containing vehicle (PBS) or TNFα (100 ng/mL) was added 24 hr later, and cells were cultured for an additional 24 hr. Samples were prepared by fixing the monolayer with 1% PFA in 1X PBS for 5 min at room temperature. Cells were washed for 5 min at room temperature with a 1X Glycine Solution to quench PFA. Plates were rinsed with ice cold 1X PBS, aspirated, and replaced with 5 mL of ice-cold Cell Scraping Solution. Cells were scraped and pelleted at 500 RCF for 10 min at 4˚C. Cell pellets were resuspended in 1 mL of Lysis Buffer and incubated on ice for 30 min. Cells were Dounce homogenized (20 strokes) to release cell nuclei. Nuclei were pelleted by centrifugation (2,400 RCF) at 4˚C for 10 min. The nuclei pellet was resuspended in 350 μL of Shearing Buffer and sonicated using a Misonix S-4000 sonicator (four 10 sec pulses with an amplitude of 50) to generate DNA fragments averaging 500 base pairs in length. Twenty micrograms of sheared chromatin were incubated with either 2 μg of IgG isotype control, anti-IKBKG, or anti-NFκB1 antibodies and incubated on an end-to-end rotator at 4˚C overnight. Samples were washed, resuspended in 50 μL of elution buffer, and incubated for 15 min on an end-to-end rotator at room temperature. Samples were reverse cross-linked by adding 50 μL of Reverse Cross-Linking Buffer to the eluted chromatin and incubated at 95˚C for 15 min. Protein was digested by adding Proteinase K (0.5 mg/mL) to each tube and incubated at 37˚C for 1 hr. Putative IKBKG and NFκB1 interactions with the *Ripk3* promoter were analyzed by quantitative PCR (qPCR). The final primer concentrations used to amplify the *Ripk3* promoter were: 250 nM (for the -0.1 kb, -0.5 kb, and -1 kb regions) and 500 nM (for the -14 kb region). Primers used for ChIP-qPCR are listed in S8 Table.

## Statistics

Results are presented as mean ± S.D. of *n* independent experiments (*n* is reported in the figure legends). Data normality was determined using the D'Agostino-Pearson omnibus test. Two-way ANOVA with a Tukey post hoc test was used to determine changes in HUVECs knocked down with nonspecific siRNA or gene of interest-specific siRNA after treatment with vehicle or TNFα or Z-VAD-FMK or NSA (Figs 2A, 2C, 2D and 3B). These analyses were chosen in order to compare two factors simultaneously: siRNAs and treatments. One-sample *t* and Wilcoxon test was used to determine fold-change of ChIP enrichment compared to the normalized negative IgG control (Fig 4A and 4B). These analyses were chosen for comparison of one variable (NFκB1 or IKBKG mean enrichment value) against the calculated IgG control enrichment level. Significance was determined by a *P*-value of 0.05 or less. All statistical analyses were achieved using GraphPad Prism8 software.

## Results

### Genomic locus proteomics (GLoPro) reveals proteins associated with the *Ripk3* locus in cultured ECs

Because cultured EC lines express varying levels of RIPK3 protein (S1 Fig), we chose an immortalized line with relatively high expression of RIPK3—the murine adult pancreatic MS1 EC line—in which to perform GLoPro and identify regulatory proteins that influence *Ripk3* transcription. We first established MS1 cells stably expressing catalytically-dead RNA-guided nuclease Cas9 (dCas9) linked to the engineered ascorbic acid peroxidase (APEX2), which can mediate rapid biotin labeling of proximal proteins in living cells. Three single guide RNAs (sgRNA) targeting the *Ripk3* locus were individually co-expressed with dCas9-APEX2 in separate lines. The sgRNAs were designed to target the following regions of the *Ripk3* locus: 39 base pairs (bp) downstream (3') of the transcription start site (TSS), 115 bp upstream (5') of the TSS, and 261 bp 5' of the TSS (respectively denoted as g39, g115, and g261, Fig 1A). Cells stably co-expressing a non-targeting sgRNA (NT-gRNA) and dCas9-APEX2 were also established as a negative control. After pre-incubation of cells with biotin-phenol and addition of hydrogen peroxide, APEX-mediated biotinylated proteins at the endothelial *Ripk3* locus were enriched and subjected to detection by liquid chromatography-mass spectrometry (LC-MS/MS).

In total, we identified 2613 proteins with g39, 2296 proteins with g115, and 2266 proteins with g261. Among these, 197 proteins were detected by all three *Ripk3*-targeting sgRNAs but were not detected with the NT-gRNA control (Fig 1B and S1 Table). Since our goal was to identify new transcriptional regulators of *Ripk3*, these 197 proteins were further filtered to select for nuclear components and transcription factors according to gene ontology (GO) term annotation, which yielded 41 proteins (Fig 1C and S2 Table). Additional analysis of GO biological process enrichment among these 41 proteins revealed that they are related to processes such as transcription and transcript processing, regulation of DNA methylation, apoptosis, and positive regulation of NF-κB signaling (S2A Fig and S3 Table). KEGG Pathway Analysis of the 41 proteins likewise highlighted transcript processing (spliceosome), apoptosis, and various cellular differentiation processes that are linked to NF-κB signaling [21,25] (S2B Fig and S4 Table).

### Identification of NFκB1 as a transcriptional regulator of *Ripk3* in ECs

RIPK3 has been linked to the NF-κB signaling pathway in non-vascular contexts [3,4], but it has not yet been identified as a transcriptional target of NF-κB signaling. Therefore, we sought to validate the transcriptional regulation of *Ripk3* by NFκB1, which was one of our 41 GLoPro hits (Fig 1C).

Since primary human umbilical vein endothelial cells (HUVECs) express low levels of RIPK3 at basal conditions (S1 Fig), we used them to explore the capacity for NFκB1 to stimulate *Ripk3* transcription. Specifically, we took advantage of the robust NF-κB signaling response that the pro-inflammatory cytokine tumor necrosis factor-alpha (TNFα) elicits in HUVECs [26]. We found that NFκB1 knockdown did not alter *Ripk3* transcription significantly in HUVECs maintained under basal conditions (Fig 2A). However, NFκB1 knockdown followed by TNFα stimulation resulted in a significant increase in *Ripk3* transcripts (Fig 2A). NFκB1 knockdown also elevated RIPK3 protein levels in HUVECs cultured with or without TNFα stimulation (Fig 2B). Importantly, we found that NFκB1 knockdown caused significant cell death in HUVECs grown under basal conditions or with TNFα stimulation, and concomitant RIPK3 knockdown rescued this cell death under both conditions (Fig 2C).

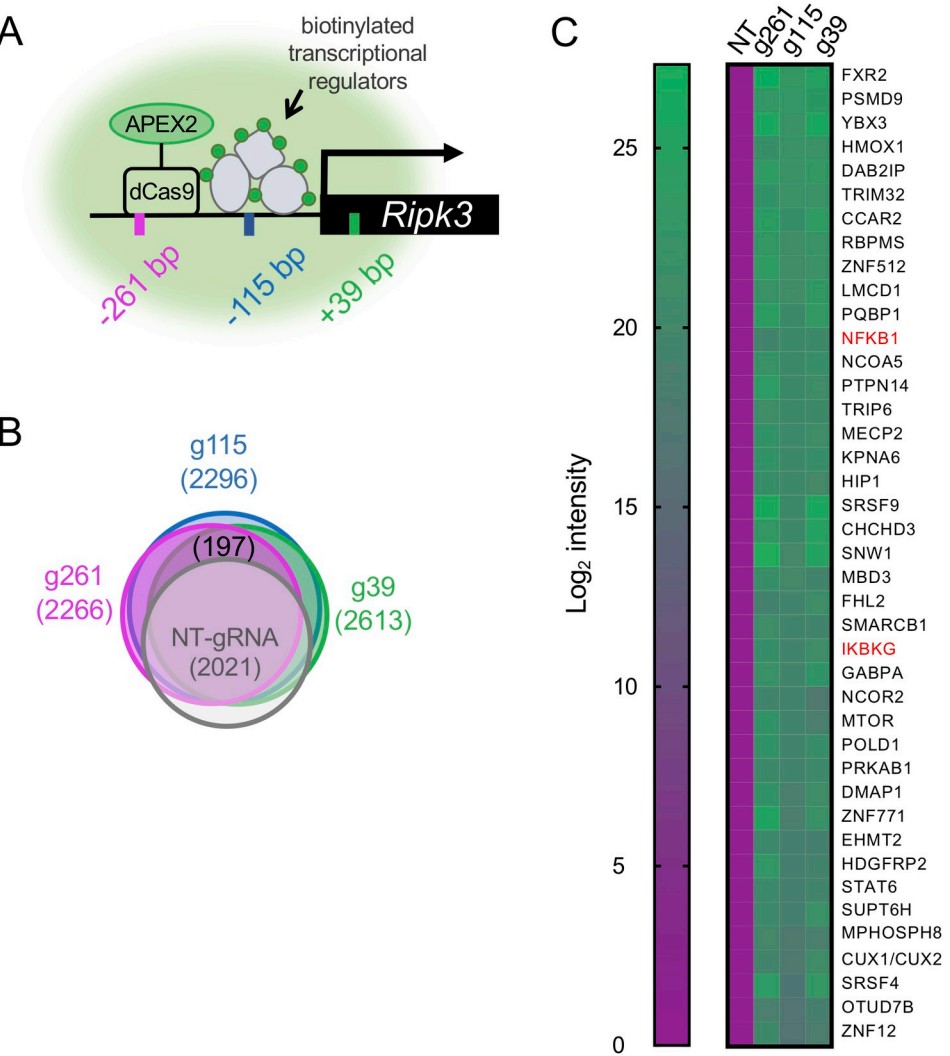

**Fig 1. Genomic locus proteomics (GLoPro) reveals proteins associated with the *Ripk3* locus in cultured ECs.** (A) Schematic of GLoPro labeling of the *Ripk3* locus in the MS1 EC line. Three targeting sgRNAs (color-coded bars) were designed to guide dCas9-APEX2 to different sequences upstream and downstream of the *Ripk3* transcription start site (black arrow). Once targeted, dCas9-APEX2 biotinylates (green circles) proximal proteins associated with the *Ripk3* locus (gray shapes), which are subsequently enriched and identified by mass spectrometry. (B) Venn diagram showing the number of proteins identified by GLoPro using the three *Ripk3*-targeting sgRNAs (denoted as g39, g115, and g261) versus a non-targeting control sgRNA (denoted as NT-gRNA). See also S1 Table. (C) Heat map of high-stringency filtered GLoPro hits arranged by Log$_2$ intensity. The 41 proteins listed were identified by all three *Ripk3*-targeting gRNAs and by annotated gene ontology (GO) terms including "nuclear as cellular component" and "transcription as molecular function." NFκB1 and IKBKG (listed in red) were further analyzed for validation as *Ripk3* transcriptional modulators (see Figs 2–4).

Moreover, the pan-caspase inhibitor Z-VAD-FMK significantly reduced the cell death observed in NFκB1 knockdown HUVECs treated with TNFα, while the necroptosis inhibitor necrosulfonamide (NSA) had no impact on this cell death (Fig 2D). These data indicate that the HUVEC death triggered by NFκB1 knockdown and TNFα treatment is mediated through a RIPK3- and caspase-dependent mechanism. Collectively, our in vitro data demonstrate that NFκB1 prevents TNFα-induced *Ripk3* transcription and subsequent cell death in primary HUVECs.

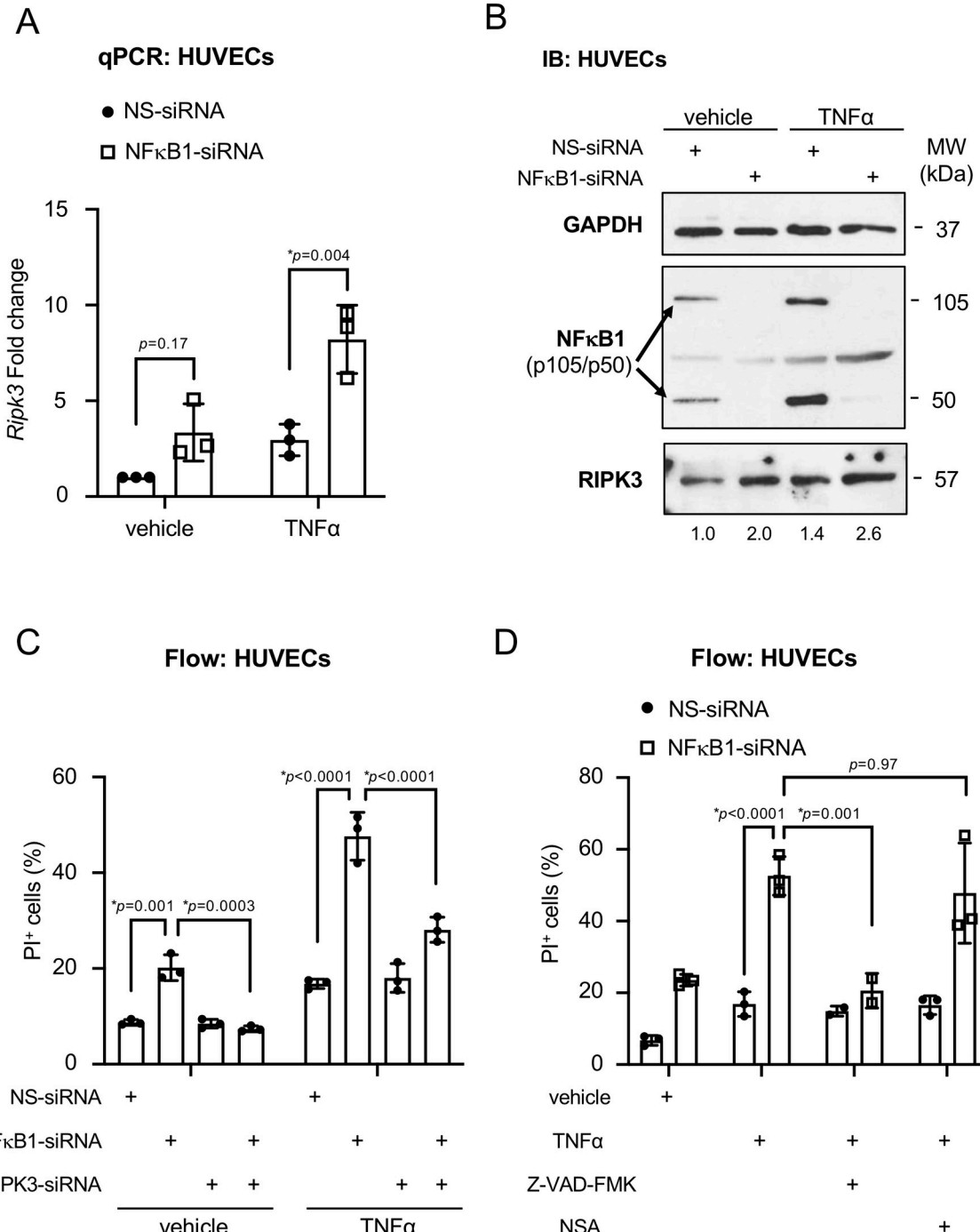

**Fig 2. NFκB1 prevents TNFα-induced *Ripk3* transcription and cell death in HUVECs.** (A, B) HUVECs were transfected with nonspecific (NS) or NFκB1-specific siRNA oligos for 24 hr and were subsequently treated with vehicle (PBS) or TNFα (100 ng/mL) for 24 hr. (A) *Ripk3* transcripts were measured by qRT-PCR. Data from 3 independent experiments were combined and are presented as relative fold change. (B) A representative immunoblot (IB) for NFκB1, RIPK3, and GAPDH is shown. Normalized densitometric values for RIPK3 are shown below the blots. (C) HUVECs were transfected with nonspecific (NS), NFκB1-specific, and/or RIPK3-specific siRNA oligos for 24 hr and were subsequently treated with vehicle (PBS) or TNFα (100 ng/mL) for 24 hr. Propidium iodide positive (PI+) cells were analyzed by flow cytometry (Flow) and were graphed as % of total cells. Data from 3 independent experiments were combined. (D) HUVECs were transfected with nonspecific (NS) or NFκB1-specific siRNA oligos for 24 hr and were subsequently treated with vehicle (PBS), TNFα (100 ng/mL), Z-VAD-FMK (50 μM), or necrosulfonamide (NSA, 5 nM) for 24 hr. PI+ cells were analyzed by flow cytometry and graphed as % of total cells. Data from 2 to 3 independent experiments were combined. All quantified data are presented as mean ± S.D. All statistics were calculated using a two-way ANOVA with a Tukey's multiple comparisons test. (*) indicates statistical significance ($p<0.05$).

## Identification of IKBKG as an additional NF-κB signaling pathway component that transcriptionally regulates *Ripk3* in ECs

In order to validate an additional target protein from our GLoPro screen (Fig 1C), we assessed the ability of IKBKG to impact *Ripk3* transcription in HUVECs. IKBKG (also known as NEMO or IKK-γ) is widely expressed and best known for its cytoplasmic roles in activating the canonical NF-κB signaling pathway and protecting cells from TNFα-induced apoptosis [27,28]. Nevertheless, IKBKG can translocate to the nucleus under genotoxic stress conditions [29], which likely accounts for its GO term annotation as "nuclear as cellular component" among our GLoPro targets (Fig 1C). We first analyzed the cellular localization of IKBKG in HUVECs by performing immunocytochemistry on cells grown under basal conditions or stimulated for 24 hr with TNFα. We detected substantial nuclear staining of IKBKG in cells grown under both conditions. These localization data supported the idea that IKBKG could play a nuclear role in regulating *Ripk3* transcription. We next analyzed *Ripk3* transcripts in HUVECs depleted of IKBKG. We did not see a significant impact on *Ripk3* transcripts when we knocked down IKBKG in HUVECs grown under basal conditions (Fig 3B). However, IKBKG knockdown followed by TNFα stimulation significantly increased *Ripk3* transcripts in HUVECs (Fig 3B), similarly to NFκB1 knockdown with TNFα stimulation (Fig 2A).

## NFκB1 and IKBKG bind to the *Ripk3* promoter in HUVECs

Finally, in order to confirm that NFκB1 and IKBKG directly bind the *Ripk3* promoter in ECs, we performed ChIP for these proteins in HUVECs. We found variable evidence for NFκB1 binding ($p = 0.14$) and significant evidence for IKBKG binding ($p = 0.03$) approximately 0.5 kb upstream (-0.5 kb) of the *Ripk3* TSS in HUVECs grown under basal conditions (Fig 4A). However, after 24 hr of TNFα treatment, we saw increased binding of both proteins at a region approximately 1 kb upstream (-1.0 kb) of the *Ripk3* TSS and significant additional binding of IKBKG closer to the TSS (-0.2 kb) (Fig 4B). Therefore, the NF-κB signaling pathway components NFκB1 and IKBKG undergo differential and increased binding to the *Ripk3* promoter in HUVECs stimulated with TNFα, which likely contributes to their suppression of *Ripk3* transcription under these conditions.

## Discussion

RIPK3 is a pleiotropic protein with known roles in necroptotic and apoptotic cell death pathways, inflammasome activation, and aerobic metabolism [30–32]. Accordingly, changes in RIPK3 expression have been associated with a range of inflammatory diseases and cancers [33]. Because RIPK3 expression can impact its function, many studies have focused on how post-translational modifications impact RIPK3 stability, particularly in the context of immune cells [34,35]. Here we sought to complement these studies by providing new insights into *Ripk3* transcriptional regulators. We designed our study to identify such regulators in ECs because our previous research indicates that endothelial RIPK3 expression levels impact vascular stability, angiogenesis, and anti-inflammatory properties [10–12]. Therefore, we reasoned that elucidation of transcriptional regulators of endothelial *Ripk3* could provide new therapeutic targets for modulating vascular diseases.

The GLoPro technique that we employed here yielded 41 proteins associated with nuclear localization and/or transcription that reside near the *Ripk3* TSS in immortalized MS1 ECs cultured under basal conditions. To address the biological relevance of this study, we utilized primary ECs to investigate the capacity of two of those proteins (NFκB1 and IKBKG) to influence *Ripk3* transcription under basal and inflammatory stimulation conditions. The other 39

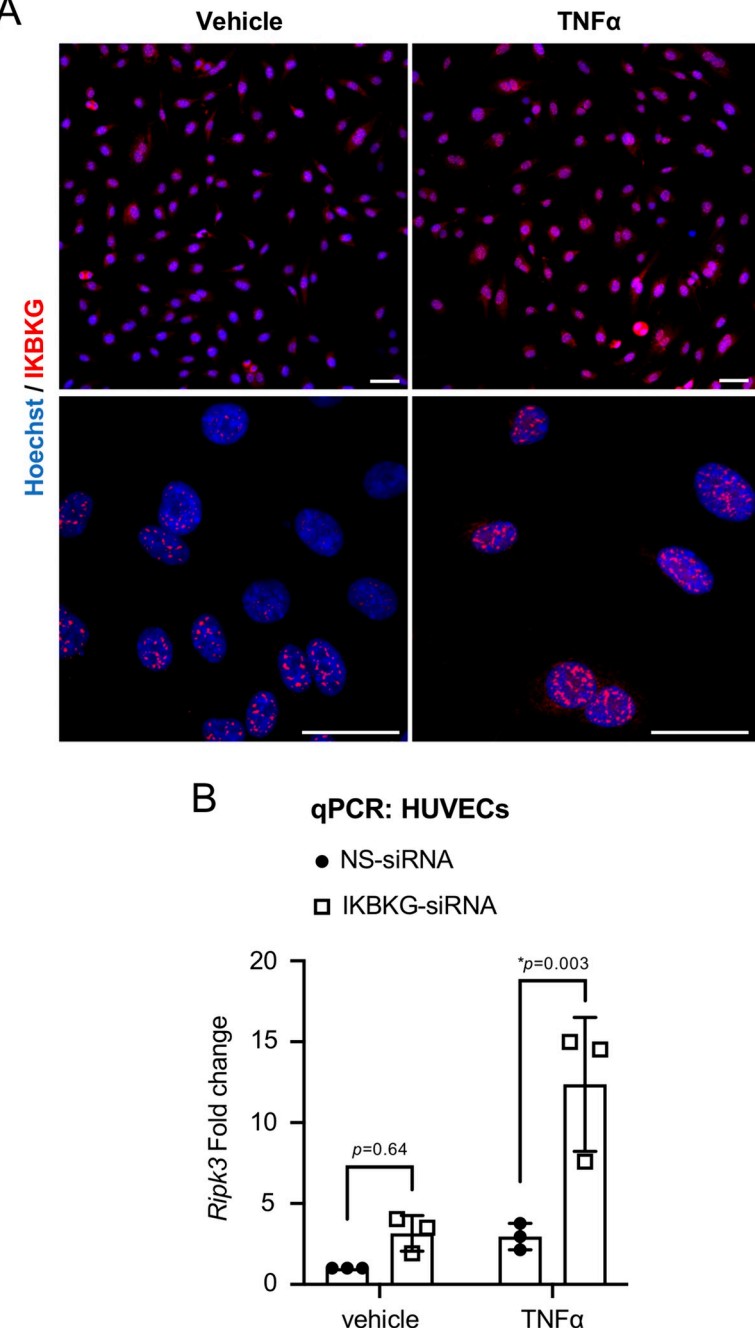

**Fig 3. IKBKG is primarily localized in the nuclei of HUVECs and suppresses *Ripk3* transcription after TNFα treatment.** (A) HUVECs were treated with vehicle or TNFα (100 ng/mL) for 24 hr and subsequently immunostained for IKBKG (red). Cellular nuclei were counterstained with Hoechst (20 μg/mL; blue). Representative images of single confocal optical sections are shown. Scale bars: 50 μm. (B) HUVECs were transfected with nonspecific (NS) or IKBKG-specific siRNA oligos for 24 hr and were subsequently treated with vehicle (PBS) or TNFα (100 ng/mL) for 24 hr. *Ripk3* transcripts were measured by qRT-PCR. Note that data generated from NS-siRNA-treated cells are identical to those shown in Fig 2A because the experiments were performed at the same time. Data from 3 independent experiments were combined and are presented as relative fold change. Data are presented as mean ± S.D. All statistics were calculated using a two-way ANOVA with a Tukey's multiple comparisons test. (*) indicates statistical significance ($p<0.05$).

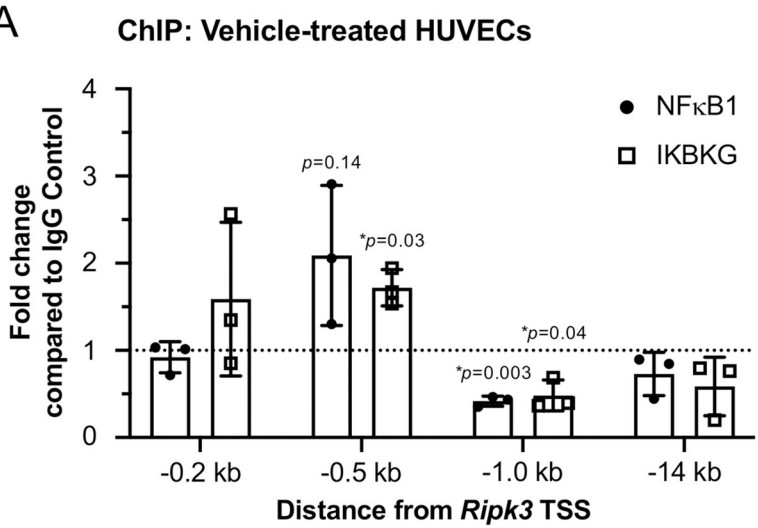

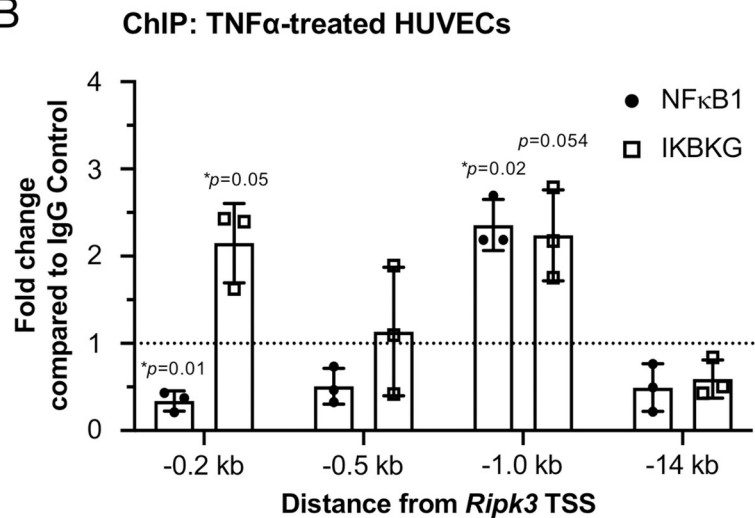

**Fig 4. NFκB1 and IKBKG bind to the *Ripk3* promoter in HUVECs.** ChIP assays using antibodies against NFκB1, IKBKG, or against rabbit IgG (as a negative control) were performed in HUVECs. Prior to ChIP, HUVECs were treated with vehicle (PBS) or TNFα (100 ng/mL) for 24 hr (A and B, respectively). Immunoprecipitated DNA was isolated and measured by qPCR to determine whether NFκB1 and IKBKG bound the *Ripk3* promoter at the regions indicated on the X-axes. Three sets of experiments were performed independently. Data are graphed as the average fold change of enrichment at the indicated sites compared to the normalized average of IgG (negative control antibody) enrichment at each of those sites (dotted line). Error bars represent S.D. All statistics were calculated using a one-sample *t* and Wilcoxon test. (*) indicates statistical significance ($p < 0.05$).

proteins that we identified remain untested and therefore provide opportunities for future examination. In addition, it would be interesting to build off our initial findings by assessing how endothelial *Ripk3* transcriptional regulators change with additional stimuli. For example, we previously reported that the chromatin remodeling enzyme CHD4 only binds to the *Ripk3* promoter and represses its transcription in embryonic ECs grown under hypoxic conditions [10]. Future GLoPro experiments performed on different subtypes of ECs or on ECs subjected to stimuli such as hypoxia, oxidatively modified lipoproteins, additional inflammatory

cytokines, and reactive oxygen species could shed new insights into the transcriptional regulation of endothelial *Ripk3* expression in various vascular disease contexts.

Multiple studies have demonstrated that RIPK3 can activate the NF-κB signaling pathway and subsequent cytokine production in monocytes, macrophages, and dendritic cells [3,4,36]. Our GLoPro screen and validation studies now demonstrate a reciprocal relationship likewise exists: NFκB1 and IKBKG can bind to the *Ripk3* promoter and suppress *Ripk3* transcription in TNFα-stimulated HUVECs. Moreover, we found that NFκB1 prevents RIPK3-mediated death of HUVECs following TNFα treatment. Since TNFα is a major inflammatory cytokine that is acutely produced when mice are challenged with the bacterial endotoxin LPS [37], it is interesting to note that mice with endothelial-specific inactivation of the NF-κB signaling pathway display LPS-induced vascular permeability and EC apoptosis [24]. However, mice exposed to more chronic, low-grade inflammation during atherosclerosis progression benefit from endothelial-specific inactivation of the NF-κB signaling pathway [38]. Therefore, we question whether TNFα dosage impacts the ability of NFκB1 and IKBKG to protect ECs from excessive endothelial *Ripk3* expression and subsequent vascular damage in vivo. If so, RIPK3 might be a beneficial therapeutic target in vascular diseases associated with high TNFα levels such as sepsis.

NFκB1 is a known suppressor of transcription when its proteolytically processed subunits (p50) homodimerize and bind to promoters of NF-κB target genes [39]. Therefore, our discovery that NFκB1 binds the *Ripk3* promoter in HUVECs and suppresses *Ripk3* transcription and cell death following TNFα stimulation, implies that *Ripk3* is directly transcriptionally repressed by p50 homodimers. However, p50 can also dimerize with other NF-κB transcription factors such as RelA, RelB, and c-Rel to promote transcription [40]. It is interesting to speculate that such heterodimers might promote *Ripk3* transcription under conditions other than those used in this study. Notably, no positive regulators of *Ripk3* transcription have yet been identified in ECs or other cell types.

Our discovery that IKBKG is also capable of binding the *Ripk3* promoter and suppressing its transcription was a true surprise for us. This protein is best known for its cytoplasmic roles in promoting NF-κB signaling by facilitating the degradation of IKK proteins that otherwise prevent NF-κB transcription factors from translocating to the cell nucleus [27]. To our knowledge, nuclear roles for IKBKG have not yet been defined in ECs, so our detection of substantial IKBKG expression in the nuclei of HUVECs grown under basal or TNFα-stimulated conditions was unexpected. Likewise, the binding of IKBKG that we saw at the *Ripk3* promoter implies that it plays a direct role in suppressing *Ripk3* transcription following TNFα stimulation. These findings help validate our GLoPro screen for *Ripk3* transcriptional regulators and pose additional questions about nuclear roles for IKBKG in ECs.

The GO terms associated with the proteins we found enriched at the *Ripk3* locus in ECs speak to other interesting regulators and functions of RIPK3. For example, in addition to its well-known roles in the necroptotic cell death pathway, RIPK3 can also regulate apoptosis of fibroblasts and epithelial cells in specific contexts [41,42]. Since our GO term analyses highlight "apoptotic process" as a biological function associated with our GLoPro-generated proteins, RIPK3 may also have apoptotic capacity in ECs to complement its documented necroptotic roles in these cells [43]. Our HUVEC studies support this hypothesis, since the pan-caspase inhibitor Z-VAD-FMK rescued NFκB1-knockdown cells treated with TNFα from death, while the necroptosis inhibitor NSA did not (Fig 2D). In addition, the GO terms highlighting "brain development", "positive regulation of neurogenesis", and "long-term memory" are interesting to us, since some studies have implicated RIPK3 in central nervous system pathologies [44,45]. This correlation highlights a need to consider endothelial RIPK3 and its regulation in the context of brain development and diseases. Finally, methylation of the

*Ripk3* promoter has been proposed to contribute to tumorigenesis [19,46,47]. We found that several protein candidates associated with the endothelial *Ripk3* locus are involved in histone and DNA methylation, including euchromatic histone-lysine N-methyltransferase 2 (EHMT2), methyl-CpG binding domain protein 3 (MBD3), and methyl CpG binding protein 2 (MECP). Future investigation will be needed to determine how these proteins regulate *Ripk3* expression in ECs and whether they impact RIPK3 function in the context of developmental and tumor angiogenesis.

In conclusion, our study provides a broad and unbiased view of transcriptional regulators found at the endothelial *Ripk3* locus and highlights the varied and complex biological processes associated with RIPK3 function. We believe these findings can serve as a foundation for further analyses of *Ripk3* transcriptional regulation and its cellular roles in both vascular and non-vascular contexts. Importantly this study also sheds new light on the relationship between the NF-κB pathway and RIPK3 at the intersection of cellular inflammation and cell death responses in ECs. Further studies will be required to assess whether NF-κB components likewise regulate *Ripk3* expression in additional cell types.

## Supporting information

**S1 Fig. (related to Results). RIPK3 protein levels vary in cultured ECs.** RIPK3 protein levels were analyzed by immunoblotting in immortalized MS1 ECs (adult murine pancreas-derived), immortalized C166 ECs (embryonic murine yolk sac-derived), and primary human umbilical vein endothelial cells (HUVECs). MS1 RIPK3 knockout (KO) ECs were generated by CRISPR/Cas9 technology and were included as a negative control. Note that the molecular weight (MW) of human RIPK3 is slightly greater than that of mouse RIPK3.
(TIF)

**S2 Fig. (related to Results). Additional annotation of filtered GLoPro hits reveals functional associations.** (A) Functional annotation clustering of GO terms associated with the 41 filtered *Ripk3* GLoPro proteins shown in Fig 1C by DAVID. (B) Kyoto Encyclopedia of Genes and Genomes (KEGG) pathway analysis of the 41 filtered *Ripk3* GLoPro proteins shown in Fig 1C. This analysis was performed using g:Profiler and included a false discovery rate of $p<0.05$ after Benjamini-Hochberg correction for multiple comparisons.
(TIF)

**S1 Table. Related to Fig 1B; 197 proteins that were detected by all three *Ripk3*-targeting sgRNAs but that were not detected with the non-targeting sgRNA control.**
(XLSX)

**S2 Table. Related to Fig 1C; Annotation of GLoPro hits encompassing nuclear transcriptional regulators.**
(XLSX)

**S3 Table. Related to Fig 1C; GO biological process enrichment analysis of the 41 proteins.**
(XLSX)

**S4 Table. Related to Fig 1C; KEGG Pathway Analysis of the 41 proteins.**
(XLSX)

**S5 Table. (related to Materials and Methods). Key resources used in this study.**
(DOCX)

**S6 Table. (related to Materials and Methods). Cloning primers used in this study.**
(DOCX)

**S7 Table. (related to Materials and Methods). qRT-PCR primers used in this study.**
(DOCX)

**S8 Table. (related to Materials and Methods). ChIP-qPCR primers used in this study.**
(DOCX)

**S9 Table. (related to Materials and Methods). Plasmids used in this study.**
(DOCX)

**S1 Raw images.**
(TIF)

**S1 References.**
(DOCX)

## Acknowledgments

We thank past and current Griffin lab members for helpful discussions. We also thank Steve Hartson (Oklahoma State University Protein Resource Core Facility) for assistance with LC-MS/MS.

## Author Contributions

**Conceptualization:** Siqi Gao.

**Funding acquisition:** Siqi Gao, Courtney T. Griffin.

**Investigation:** Siqi Gao, Matthew Menendez, Katarzyna Kurylowicz.

**Writing – original draft:** Siqi Gao.

**Writing – review & editing:** Siqi Gao, Matthew Menendez, Courtney T. Griffin.

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
