## [Decision Letter · Decision Letter 0]

10 Dec 2020

PONE-D-20-35540

Genomic locus proteomic screening identifies NF k B1 as a transcriptional regulator of Ripk3 in endothelial cells

PLOS ONE

Dr. Griffin,

Thank you for submitting your manuscript to PLOS ONE. After careful consideration, we feel that it has merit but does not fully meet PLOS ONE’s publication criteria as it currently stands. Therefore, we invite you to submit a revised version of the manuscript that addresses the points raised during the review process.

Some of the changes necessary for acceptance include: additional validation of targets obtained from the GloPro assay, addressing sensitivity of the assays [GloPro and the low RIPK3 protein levels in endothelial cells (ECs)] and the need for rigorous quantification of all data in the paper. In addition, all comments related to localization of RIPK3, and its functional relevance with NF-kb binding to its promoter, and the functional role of RIPK3 in ECs must be constructively addressed. The authors are also encouraged to provide additional bioinformatic and mutational data for the NF-kb binding sites on the promoter, as requested by the reviewer but this is not necessary for acceptance.  

Please submit your revised manuscript by March 8, 2021. However, if you will need more time than this to complete your revisions, please reply to this message or contact the journal office at plosone@plos.org. Please include the following items when submitting your revised manuscript:

We look forward to receiving your revised manuscript.

Kind regards,

Ramani Ramchandran

Academic Editor

PLOS ONE

Journal Requirements:

Reviewers' comments:

Reviewer's Responses to Questions

**Comments to the Author**

1. Is the manuscript technically sound, and do the data support the conclusions?

Reviewer #1: Yes

Reviewer #2: Partly

Reviewer #3: Yes

2. Has the statistical analysis been performed appropriately and rigorously? 

Reviewer #1: Yes

Reviewer #2: Yes

Reviewer #3: Yes

3. Have the authors made all data underlying the findings in their manuscript fully available?

Reviewer #1: Yes

Reviewer #2: Yes

Reviewer #3: Yes

4. Is the manuscript presented in an intelligible fashion and written in standard English?

Reviewer #1: Yes

Reviewer #2: No

Reviewer #3: Yes

5. Review Comments to the Author

Reviewer #1: Although the manuscript was written well. There are few concerns to consider. Please see below for the clarifications.

1. RIPK3 is predominantly in the cytoplasm, by any means did the author assess the interaction is cytoplasmic or nuclear.

2. It is more concerning that if NFkB1 knockdown caused significant cell death in HUVECs grown under basal conditions or with TNF-a stimulation, when was the transcriptional and translational experiments done.

3. It implies that TNF-a only have transcriptional regulation of RIPK3 but not translational.

4. What is the rationale for using pan-caspase inhibitor Z-VAD-FMK to rescue NF-kB knockdown mediated cell death in HUVECs? It appears too vague.

5. If endothelial RIPK3 expression must be tightly regulated to maintain vascular integrity and homeostasis at different stages of life, the low-level expression of RIPK3 in HUVECs suggest HUVECs may not be the right model. Could you please clarify the rationale for using HUVECs?

6. In the discussion section, many studies have focused on how post-translational modifications impact RIPK3 stability, particularly in the context of immune cells, could you please give any references to that.

7. In the discussion section, according to the author, the study was designed to identify regulators such as RIPK3 in ECs, is HUVECs the right model to study such kind of regulator. The rationale is not clear.

8. In the discussion section some of the information about NF-kB can be moved to introduction.

Reviewer #2: Gao et al., have investigated, “Genomic locus proteomic screening identifies NFB1 as 1 a transcriptional regulator of Ripk3 in endothelial cells”. Authors have screened transcription factors and coregulatory proteins associated Ripk3 using GLoPro – CRISPR approach in MS1 and HUVEC cells. There are few issues that need to be addressed before publication as follows:

General:

1. The results section is poorly written and has not explained in depth to support their experimental conclusions.

2. Figure2, need better color contrast to observe heat map matrix. Secondly, Merge Figure 1 and 2.

3. Figure 3 panel A&B should be moved to Supplementary. It does not add any experimental parameters for the manuscript.

Specific Comments:

4. GLoPro – CRISPR approach/ screening is the backbone of the manuscript. Authors should validate more identified proteins as are listed in Figure 2. Simple heatmap is not enough for being a main manuscript Figure.

5. Supporting Figure S1 show that HUVEC cells have very low protein expression (~negligible) of RIPK3 as compared to MS1 cells, however, CHIP -IP assay, lane NFkB1 in Figure 4 panel A&B, do not co-relate. It is very important to have quantitative enrichment or fold change for each test samples with some Statistically Analysis.

6. Figure 5 panel B need quantification as well as more detail in result section. Why NFkB1 (p105/p50) is high? What are other bands at MW 105 and 50?

Reviewer #3: Gao et al perform Genomic Locus Proteome (GLoPro) screening to identify proteins that interact with the RIPK3 promoter in endothelial cells. By utilizing 3 sgRNA probes, they identify a number of overlapping proteins, including 41 nuclear proteins, some of which are transcription factors. One of these transcription factors, NF-kB1, binds the RIPK3 promoter in HUVEC by ChIP and inhibits RIPK3 expression in the presence of TNF-alpha. They furthermore show that knock-down of NF-kB1 induces cell death in a RIPK3-dependent manner. This is an interesting approach to identify potential regulators, but there are several limitations.

Major Comments:

1) What is the resolution of GLoPro? Would you anticipate that all three probes would identify the same interacting proteins? For the MS1 NF-kB1 ChIP data, it seems to only interact with a distal region (-1 kb), but the GLoPro showed interaction with all three probes, which were more proximal to the promoter. Is this due to differences in resolution between these two techniques?

2) Are there cis elements in the RIPK3 promoter that match the transcription factors that were identified? This is especially relevant for NF-kB1. Where are the NF-kB elements and is the binding predicted to be conserved across species? Does mutation of NF-kB motifs affect promoter activity in a luciferase assay?

3) It would be helpful if there was additional validation of the GLoPro technique beyond just NF-kB1.

4) Does NF-kB1 knock-down in MS1 cells affect RIPK3 expression?

Minor Comments:

1) It isn’t clear how the control probe was designed.

2) Are the GO terms shown in the figures complete or selected GO terms? This should be indicated in the figure legends.

3) Information on the number of replicates that were performed should be included in the figure legends.

4) In Figure 2, it would be helpful to show the data from the control oligo in the heat map for comparison purposes.

6. PLOS authors have the option to publish the peer review history of their article (what does this mean?). If published, this will include your full peer review and any attached files.

Reviewer #1: **Yes: **KARTHIKEYAN THIRUGNANAM

Reviewer #2: No

Reviewer #3: No

---

## [Author Response · Author response to Decision Letter 0]

4 May 2021

Please see the attached "Response to Reviewers" for our full response (including figures)

April 28, 2021

Ramani Ramchandran

Academic Editor

PLOS ONE

RE: Resubmission of PONE-D-20-35540

Dear Dr. Ramchandran:

Thank you for handling the recent reviews of our manuscript “Genomic locus proteomic screening identifies NF�B1 as a transcriptional regulator of Ripk3 in endothelial cells.” We appreciated the Re-viewers’ constructive critiques and have generated new data and re-written/organized sections of our original manuscript to address their queries, as detailed below.

Reviewer #1

1. “RIPK3 is predominantly in the cytoplasm, by any means did the author assess the interaction is cytoplasmic or nuclear.”

We agree with the Reviewer that RIPK3 is predominantly a cytoplasmic protein in cell types that have been assessed for its localization. Because the goal of this manuscript was to identify tran-scriptional regulators of Ripk3, the genomic locus proteomic (GLoPro) screen that we utilized here was performed to identify proteins that interact with the Ripk3 promoter—not with RIPK3 protein itself. Nevertheless, we recently performed an unbiased proteomic screen to identify RIPK3-interacting proteins in both HUVECs and the MS1 endothelial cell line [1]. NF�B1 and IKBKG (newly analyzed in this revision) did not emerge in our mass spectrometry analysis of proteins that immunoprecipitated with RIPK3 in these endothelial cell lines. Therefore, our collec-tive data indicate that NF�B1 and IKBKG interact with the Ripk3 promoter in endothelial cells but not with RIPK3 protein itself.

2. “It is more concerning that if NF�B1 knockdown caused significant cell death in HUVECs grown under basal conditions or with TNF� stimulation, when was the transcriptional and translational experiments done?”

RIPK3 transcriptional and translational experiments (original Figures 5A and 5B; revised Figures 2A and 2B, respectively) were performed at 48 hr (24 hr after NF�B1 knockdown plus 24 hr after additional vehicle or TNF� treatment). This is comparable to the timecourse of analysis for cell death (original Figures 5C and 5D; revised Figures 2C and 2D). Please note that we routinely rinse cultured cells before all analyses to remove dead cells and debris. Moreover, transcription is evaluated by qPCR using three housekeeping genes as internal controls, and GAPDH is used as a loading control for our western blots. Therefore, we are confident that we are measuring Ripk3 transcripts and protein in live cells rather than dead ones.

3. “It implies that TNF� only have transcriptional regulation of RIPK3 but not translational.”

TNF� treatment has a small impact on Ripk3 transcription (original Figure 5A; revised Figure 2A, lanes 1 vs. 3: p=0.28) and translation (original Figure 5B; revised Figure 2A, lanes 1 vs. 3), but a bigger impact when combined with NF�B1 knockdown (original Figure 5A and 5B; revised Fig-ures 2A and 2B, lanes 1 vs. 4). Admittedly, the transcriptional impact of this combination is great-er than the translational impact (8.2-fold transcriptional change vs. 2.6-fold translational change). Nevertheless, the functional relevance of TNF� treatment plus NF�B1 knockdown is evident from the cell death assays (original Figure 5C; revised Figure 2C, lane 6) (original Figure 5D; re-vised Figure 2D, lane 4).

4. “What is the rationale for using pan-caspase inhibitor Z-VAD-FMK to rescue NF�B1 knockdown mediated cell death in HUVECS? It appears too vague.”

Z-VAD-FMK is a pan-caspase inhibitor that has been used since the mid-1990s to block pro-grammed cell death in a variety of mammalian cell lines [2]. It is commonly employed to distin-guish apoptosis from non-caspase-mediated forms of cell death such as necroptosis [3]. Z-VAD-FMK is also frequently used in combination with TNF� stimulation to block apoptosis and drive cultured cells toward RIPK3-dependent necroptosis [4, 5]. When we determined that the elevated cell death we detected in HUVECs treated with TNF� and NF�B1 knockdown was RIPK3-dependent (original Figure 5C; revised Figure 2C, lanes 6 vs. 8), we assumed the death would be mechanistically necroptotic rather than apoptotic. Therefore, we predicted that we would be able to block the cell death with the necroptosis inhibitor necrosulfonamide (NSA) but not with the apoptosis inhibitor Z-VAD-FMK. However, our cell death blocking experiments gave the opposite results (original Figure 5D; revised Figure 2D), indicating that the RIPK3-mediated death we stimulated in HUVECs was apoptotic and/or caspase-dependent rather than necroptotic. This result adds to our lab’s growing evidence that RIPK3-mediated necroptosis is difficult to stimulate or detect in endothelial cells both in vitro and in vivo [6, 7].

5. “If endothelial RIPK3 expression must be tightly regulated to maintain vascular integrity and ho-meostasis at different stages of life, the low-level expression of RIPK3 in HUVECs suggest HU-VECs may not be the right model. Could you please clarify the rationale for using HUVECs?”

Although the immortalized MS1 endothelial cell line we used for the GLoPro studies was critical for generating the large number of cells required for this assay, we believe that HUVECs are more appropriate for functional validation studies since they are primary cells and are maintained at low passage number (endothelial cells are notoriously plastic and prone to de-differentiation in prolonged culture[8]). Notably, HUVECs are responsive to TNF�-mediated NF-�B activation and downstream signaling events [9], which makes them particularly relevant to this study. In addi-tion, we recently used HUVECs to study novel angiogenic roles for RIPK3, and they faithfully re-capitulated endothelial cell behaviors that occurred upon Ripk3 genetic deletion in vivo [1]. There-fore, although RIPK3 is expressed at relatively low levels in unstimulated HUVECs compared to immortalized endothelial cells (see Figure S1), RIPK3 is clearly functional and biologically rele-vant in HUVECs and requires regulated maintenance.

6. “In the discussion section, many studies have focused on how post-translational modifications impact RIPK3 stability, particularly in the context of immune cells, could you please give any ref-erences to that.”

We have added relevant references, as requested (see new references #34, 35). Please see al-so references #13-16, which were previously cited in our Introduction.

7. “In the discussion section, according to the author, the study was designed to identify regulators such as RIPK3 in ECs, is HUVEC the right model to study such kind of regulator. The rationale is not clear.”

Please see our response to #5 above.

8. “In the discussion section some of the information about NF-�B can be moved to introduction.”

We have moved background information about NF-�B signaling in ECs to the introduction, as suggested.

Reviewer #2

1. “The results section is poorly written and has not explained in depth to support their experimental conclusions.”

We have revised the Results section to clarify the techniques and motivations behind all the ex-periments.

2. “Figure 2 need better color contrast to observe heat map matrix. Secondly, merge Figure 1 and 2.”

We have revised and merged Figures 1 and 2, as suggested.

3. “Figure 3 should be moved to Supplementary. It does not add any experimental parameters for the manuscript.”

The original Figure 3 is now Figure S2, as suggested. 

4. “GLoPro-CRISPR approach/screening is the backbone of the manuscript. Authors should vali-date more identified proteins as are listed in Figure 2.”

We appreciated this important recommendation and have now evaluated IKBKG (also known as NEMO or IKK-�) for its ability to bind the Ripk3 promoter and regulate its transcription. We chose this protein among those gleaned from the GLoPro screen because of its important role in the NF-�B signaling pathway, which is a critical component of our manuscript. Notably, IKBKG is best known for its cytoplasmic roles in activating the NF-�B pathway and in protecting cells from TNF�-induced apoptosis [10, 11]. However, IKBKG can also undergo nuclear translocation upon stress induction in a pre-B cell line [12]. Interestingly, we now show that IKBKG is primarily ex-pressed in the nuclei of HUVECs grown under basal conditions, and its nuclear expression ap-pears to be further elevated upon TNF� treatment (see revised Figure 3A). Importantly, knock-down of IKBKG significantly elevates Ripk3 transcription in TNF�-treated HUVECs (see revised Figure 3B, lane 4)—similarly to the Ripk3 elevation seen with NF�B1 knockdown plus TNF� treatment (revised Figure 2A, lane 4)—which indicates that both of these NF-�B pathway com-ponents suppress endothelial Ripk3 transcription upon TNF� treatment. Importantly, our ability to ChIP IKBKG to the Ripk3 promoter in HUVECs grown under basal conditions (revised Figure 4A, lane 4) and to detect its recruitment to the -0.2 kb and -1.0 kb sites of the promoter after TNF� treatment (revised Figure 4B, lanes 2 and 6) support our conclusion that IKBKG directly regu-lates Ripk3 transcription in HUVECs, particularly after TNF� treatment. These new ChIP, im-munostaining, and transcriptional data further validate our GLoPro approach while also providing novel insights into nuclear roles for IKBKG in endothelial cells.

5. “Supporting Figure S1 show that HUVEC cells have very low protein expression (~negligible) of RIPK3 as compared to MS1 cells, however, ChIP assay, lane NF�B1 in Figure 4 panel A & B, do not co-relate. It is very important to have quantitative enrichment or fold change for each test samples with some Statistically Analysis.”

To address this important request, we repeated all our HUVEC ChIP analyses in triplicate and have analyzed the data by qPCR for quantitative analysis of NF�B1 and IKBKG enrichment at the Ripk3 promoter (see revised Figure 4). Also, please see responses to Reviewer 1 (#3 and #5) about Ripk3 transcript vs. protein levels in HUVECs

6. “Figure 5 panel B need quantification as well as more detail in result section. Why NFkB1 (p105/p50) is high? What are other bands at MW 105 and 50?”

The original Figure 5B is now revised Figure 2B. Please note that quantification (relative densi-tometry of RIPK3) is provided below the RIPK3 blot. For the NF�B1 blot, the antibody used rec-ognizes both subunits of the protein (p105 and p50), which are shown on the blot at their corre-sponding molecular weights. We have now added arrows to clarify that those are the bands of in-terest (the band located between 105 and 50 kDa is a background band picked up by the anti-body).

Reviewer #3

1. (Major) “What is the resolution of GLoPro? Would you anticipate that all three probes would iden-tify the same interacting proteins? For the MS1 NF�B1 ChIP data, it seems to only interact with a distal region (-1 kb), but the GLoPro showed interaction with all three probes, which were more proximal to the promoter. Is this due to differences in resolution between these two techniques?”

Myers et al reported that the GLoPro technique biotinylates proteins within 400bp of a genomic target [13]. Therefore, given that our sgRNAs spanned a region of 300 bp (-261 through +39 of the Ripk3 promoter), we do predict that they would identify a largely overlapping set of interacting proteins. Our original ChIP for NF�B1 binding in MS1 cells showed the protein sitting ~1 kb up-stream (-1.0 kb) of the Ripk3 transcription start site under basal conditions (original Figure 4A, also shown in Figure R1D below). For this revision we performed more extensive ChIP analysis of HUVECs and found NF�B1 and IKBKG sitting ~0.5 kb upstream of the TSS under basal condi-tions (p=0.14 and 0.03, respectively; see revised Figure 4A). Because the majority of our soni-cation fragments for ChIP are ~250-750 bp in size, we believe our ChIP and GLoPro data are consistent with each other in terms of indicating that NF�B1 and IKBKG can bind within 1 kb of the Ripk3 TSS in endothelial cells. 

2. (Major) “Are there cis elements in the RIPK3 promoter that match the transcription factors that were identified? This is especially relevant for NF�B1. Where are the NF-�B elements and is the binding predicted to be conserved across species? Does mutation of NF-�B motifs affect pro-moter activity in a luciferase assay?”

We used the Meme Suite (https://meme-suite.org/meme/) to analyze the murine Ripk3 promoter for predicted NF-�B transcription factor binding sites. The position weight matrix shown below in Figure R1 demonstrates an NF-�B consensus binding motif located between -1258 bp and -1237 bp within the Ripk3 promoter. This binding motif is based on the described consensus DNA se-quence of 5’-GGGRNYYYCC-3’ (in which R is a purine, Y is a pyrimidine, and N is any nucleo-tide) that has been reported for NF-�B transcription factors [14]. Although the Meme Suite did not predict a comparable NF-�B binding site in the human Ripk3 promoter, our lab has shown that there is high sequence conservation of the Ripk3 promoter around the -1.0 kb and -0.2 kb re-gions across species [6]. This conservation guided our analysis of these regions in our ChIP ex-periments. Please note that because we could not pinpoint specific NF�B1 or IKBKG binding se-quences in the human Ripk3 promoter, we could not perform the relevant luciferase assays suggested above in HUVECs.

3. (Major) “It would be helpful if there was additional validation of the GLoPro technique beyond just NF�B1.”

Please see our response to Reviewer 2, question #4.

4. (Major) “Does NF�B1 knock-down in MS1 cells affect RIPK3 expression?”

We used the immortalized MS1 endothelial cell line to perform GLoPro because this assay re-quires clonal selection and a large number of cells (~4X108), which are impractical hurdles when using primary endothelial cells like HUVECs. When we obtained our GLoPro results, we initially began validating them in the MS1 line. Indeed, we did see a small but significant decrease in Ripk3 transcripts when we knocked down NF�B1 (see Figure R2A below). We likewise saw a small but significant decrease in luciferase activity when we knocked down NF�B1 in MS1 cells transfected with a luciferase reporter driven by a portion of the Ripk3 promoter to which we can ChIP NF�B1 (see Figure R2B,D below). Finally, we saw a small but significant increase in Ripk3 transcripts when we overexpressed NF�B1 in MS1 cells (see Figure R2C below). Together these data indicate that NF�B1 manipulation can impact Ripk3 transcription in MS1 cells, but be-cause this is an immortalized cell line, we thought it would be important to validate the ability of NF�B1 to influence Ripk3 transcription in primary endothelial cells such as HUVECs (see re-sponse to Reviewer #1, question #5). Our HUVEC data do validate NF�B1 as a transcriptional regulator of Ripk3, but they reveal a different regulatory relationship from what we saw in MS1 cells: NF�B1 represses Ripk3 transcription when HUVECs are challenged with TNF�—a major stimulus for NF-�B signaling. We believe the large and highly significant effects we see on Ripk3 transcription and on RIPK3-mediated cell death in HUVECs following NF�B1 knockdown and TNF� stimulation are more compelling than the nominal changes we saw in baseline MS1 cells with NF�B1 manipulation.

In summary, we acknowledge the different effects on Ripk3 transcription when we knockdown NF�B1 in MS1 cells (small reduction) versus HUVECs (big increase when cells are stimulated with TNF�). We also acknowledge that baseline RIPK3 protein is much more highly expressed in the two immortalized endothelial cell lines we analyzed (MS1 and C166) than in HUVECs (original Figure S1). We don’t know the reason for this expression difference, but we suspect the high levels of RIPK3 in MS1 cells made it beneficial for using GLoPro to identify proteins that sit at the Ripk3 promoter and impact its transcription. Nevertheless, we maintain that functional validation of the true regulatory roles of our GLoPro-identified proteins is best reserved for primary endo-thelial cells like HUVECs with more physiological RIPK3 expression.* 

*We have only seen substantial in vivo endothelial RIPK3 expression in mouse embryos dur-ing stages of active angiogenesis [1, 6, 7].

5. (Minor) “It isn’t clear how the control probe was designed.”

The control probe for the GLoPro analysis (non-targeting gRNA; NT-gRNA) was designed so as not to recognize any sequence in the mouse genome, as described [15].

6. (Minor) “Are the GO terms shown in the figures complete or selected GO terms? This should be indicated in the figure legends.”

The GO terms associated with the GLoPro heat map (original Figure 2; revised Figure S2A) were filtered to include terms that include “nuclear as cellular component” and “transcription as molecular function,” since our goal was to identify transcriptional regulators of Ripk3. The GO terms shown in the original Figure 3 (revised Figure S2A) are further (unfiltered) functional anno-tations of the 41 “nuclear” proteins identified in Figure 2. These distinctions are explained in the legend for Figure S2.

7. (Minor) “Information on the number of replicates that were performed should be included in the figure legends.”

This information is now included in the figure legends. 

8. (Minor) “In Figure 2, it would be helpful to show the data from the control oligo in the heat map for comparison purposes.”

We have modified the heat map (now shown in revised Figure 1C) to accommodate this sugges-tion.

In summary, we have made the following additions and changes to this manuscript in response to the Reviewers’ suggestions:

• Combined original Figures 1 and 2

• Moved original Figure 3 to the supplement (now Figure S2)

• Generated new analysis of an additional GLoPro target protein and NF-�B signaling pathway component: IKBKG. We showed by immunostaining that IKBKG has significant nuclear locali-zation in HUVECs grown under basal conditions or with TNF� stimulation (new Figure 3A). We also showed knockdown of IKBKG in HUVECs stimulated with TNF� significantly ele-vates Ripk3 transcripts (new Figure 3B)*. 

*Please note that while analyzing Ripk3 transcription following IKBKG knockdown, we also repeated NF�B1 knockdown in the same experimental runs (both as a control and as fur-ther validation of the NF�B1 effects on Ripk3 transcription from a different author). Our revised Figure 2A now includes our most current analyses of NF�B1 knockdown effects on Ripk3 transcription. Note, the data from non-specific-siRNA-treated cells are identical for Figure 2A and 3B, since the analyses were performed at the same time (this is indicat-ed in our new figure legend for Figure 3). Therefore, the effects of NF�B1 and IKBKG knockdown can be directly compared because they were generated in the same experi-ments, but we separated out the data into Figures 2A and 3B so that we could present the NF�B1 and IKBKG stories sequentially and logically.

• Generated all new ChIP-qPCR analysis of NF�B1 and IKBKG binding to the Ripk3 promoter in HUVECs, both at basal conditions and after TNF� treatment (revised Figure 4). ChIP ex-periments were performed three separate times so that appropriate quantification and statis-tical analyses of the findings could be presented, as requested.

• Manuscript text was modified to include the new IKBKG data and to incorporate Reviewer suggestions.

Thank you for considering these revisions; we believe they shed additional novel insights into endo-thelial Ripk3 transcriptional regulation, particularly under inflammatory stimulation.

Sincerely,

Courtney Griffin, Ph.D.

References for Rebuttal Letter

1. Gao S, Griffin CT. RIPK3 modulates growth factor receptor expression in endothelial cells to support angiogenesis. Angiogenesis. 2021. Epub 2021/01/16. doi: 10.1007/s10456-020-09763-5. PubMed PMID: 33449298.

2. Jacobsen MD, Weil M, Raff MC. Role of Ced-3/ICE-family proteases in staurosporine-induced programmed cell death. J Cell Biol. 1996;133(5):1041-51. Epub 1996/06/01. doi: 10.1083/jcb.133.5.1041. PubMed PMID: 8655577; PubMed Central PMCID: PMCPMC2120856.

3. Galluzzi L, Kepp O, Kroemer G. RIP kinases initiate programmed necrosis. J Mol Cell Biol. 2009;1(1):8-10. Epub 2009/08/15. doi: 10.1093/jmcb/mjp007. PubMed PMID: 19679643.

4. Narayan N, Lee IH, Borenstein R, Sun J, Wong R, Tong G, et al. The NAD-dependent deacetylase SIRT2 is required for programmed necrosis. Nature. 2012;492(7428):199-204. Epub 2012/12/04. doi: 10.1038/nature11700. PubMed PMID: 23201684.

5. Degterev A, Zhou W, Maki JL, Yuan J. Assays for necroptosis and activity of RIP kinases. Methods Enzymol. 2014;545:1-33. doi: 10.1016/B978-0-12-801430-1.00001-9. PubMed PMID: 25065884.

6. Colijn S, Gao S, Ingram KG, Menendez M, Muthukumar V, Silasi-Mansat R, et al. The NuRD chromatin-remodeling complex enzyme CHD4 prevents hypoxia-induced endothelial Ripk3 transcription and murine embryonic vascular rupture. Cell Death Differ. 2020;27(2):618-31. Epub 2019/06/27. doi: 10.1038/s41418-019-0376-8. PubMed PMID: 31235857.

7. Colijn S, Muthukumar V, Xie J, Gao S, Griffin CT. Cell-specific and athero-protective roles for RIPK3 in a murine model of atherosclerosis. Dis Model Mech. 2020;13(1). Epub 2020/01/19. doi: 10.1242/dmm.041962. PubMed PMID: 31953345; PubMed Central PMCID: PMCPMC6994951.

8. Lacorre DA, Baekkevold ES, Garrido I, Brandtzaeg P, Haraldsen G, Amalric F, et al. Plasticity of endothelial cells: rapid dedifferentiation of freshly isolated high endothelial venule endothelial cells outside the lymphoid tissue microenvironment. Blood. 2004;103(11):4164-72. Epub 2004/02/21. doi: 10.1182/blood-2003-10-3537. PubMed PMID: 14976058.

9. Xia P, Gamble JR, Rye KA, Wang L, Hii CS, Cockerill P, et al. Tumor necrosis factor-alpha induces adhesion molecule expression through the sphingosine kinase pathway. Proc Natl Acad Sci U S A. 1998;95(24):14196-201. Epub 1998/11/25. doi: 10.1073/pnas.95.24.14196. PubMed PMID: 9826677; PubMed Central PMCID: PMCPMC24350.

10. Israel A. The IKK complex, a central regulator of NF-kappaB activation. Cold Spring Harb Perspect Biol. 2010;2(3):a000158. Epub 2010/03/20. doi: 10.1101/cshperspect.a000158. PubMed PMID: 20300203; PubMed Central PMCID: PMCPMC2829958.

11. Legarda-Addison D, Hase H, O'Donnell MA, Ting AT. NEMO/IKKgamma regulates an early NF-kappaB-independent cell-death checkpoint during TNF signaling. Cell Death Differ. 2009;16(9):1279-88. Epub 2009/04/18. doi: 10.1038/cdd.2009.41. PubMed PMID: 19373245; PubMed Central PMCID: PMCPMC2728158.

12. Hwang B, McCool K, Wan J, Wuerzberger-Davis SM, Young EW, Choi EY, et al. IPO3-mediated Nonclassical Nuclear Import of NF-kappaB Essential Modulator (NEMO) Drives DNA Damage-dependent NF-kappaB Activation. J Biol Chem. 2015;290(29):17967-84. Epub 2015/06/11. doi: 10.1074/jbc.M115.645960. PubMed PMID: 26060253; PubMed Central PMCID: PMCPMC4505044.

13. Myers SA, Wright J, Peckner R, Kalish BT, Zhang F, Carr SA. Discovery of proteins associated with a predefined genomic locus via dCas9-APEX-mediated proximity labeling. Nat Methods. 2018;15(6):437-9. Epub 2018/05/08. doi: 10.1038/s41592-018-0007-1. PubMed PMID: 29735997; PubMed Central PMCID: PMCPMC6202184.

14. Wan F, Lenardo MJ. Specification of DNA binding activity of NF-kappaB proteins. Cold Spring Harb Perspect Biol. 2009;1(4):a000067. Epub 2010/01/13. doi: 10.1101/cshperspect.a000067. PubMed PMID: 20066093; PubMed Central PMCID: PMCPMC2773628.

15. Doench JG, Fusi N, Sullender M, Hegde M, Vaimberg EW, Donovan KF, et al. Optimized sgRNA design to maximize activity and minimize off-target effects of CRISPR-Cas9. Nat Biotechnol. 2016;34(2):184-91. Epub 2016/01/19. doi: 10.1038/nbt.3437. PubMed PMID: 26780180; PubMed Central PMCID: PMCPMC4744125.

---

## [Decision Letter · Decision Letter 1]

25 May 2021

PONE-D-20-35540R1

Genomic locus proteomic screening identifies the NF-kB signaling pathway components NFkB1 and IKBKG as transcriptional regulators of Ripk3 in endothelial cells

PLOS ONE

Dear Dr. Griffin,

Thank you for submitting your manuscript to PLOS ONE. After careful consideration, we feel that it has merit but does not fully meet PLOS ONE’s publication criteria as it currently stands. Therefore, we invite you to submit a revised version of the manuscript that addresses the points raised during the review process.

Please address the statistical comments raised by one of the reviewers. Statistical rigor is part of our publication criteria at PLoS ONE.

Please submit your revised manuscript by June 15, 2021. Please include the following items when submitting your revised manuscript:

We look forward to receiving your revised manuscript.

Kind regards,

Ramani Ramchandran

Academic Editor

PLOS ONE

Journal Requirements:

Reviewers' comments:

Reviewer's Responses to Questions

**Comments to the Author**

1. If the authors have adequately addressed your comments raised in a previous round of review and you feel that this manuscript is now acceptable for publication, you may indicate that here to bypass the “Comments to the Author” section, enter your conflict of interest statement in the “Confidential to Editor” section, and submit your "Accept" recommendation.

Reviewer #1: All comments have been addressed

Reviewer #2: All comments have been addressed

Reviewer #3: All comments have been addressed

2. Is the manuscript technically sound, and do the data support the conclusions?

Reviewer #1: Yes

Reviewer #2: Yes

Reviewer #3: Yes

3. Has the statistical analysis been performed appropriately and rigorously? 

Reviewer #1: Yes

Reviewer #2: Yes

Reviewer #3: Yes

4. Have the authors made all data underlying the findings in their manuscript fully available?

Reviewer #1: Yes

Reviewer #2: (No Response)

Reviewer #3: Yes

5. Is the manuscript presented in an intelligible fashion and written in standard English?

Reviewer #1: Yes

Reviewer #2: Yes

Reviewer #3: Yes

6. Review Comments to the Author

Reviewer #1: I thank the authors for addressing all the comments and making the manuscript more commendable. I would also recommend studying RIPK3 in other primary endothelial cells where RIPK3 is expressing in high levels.

Reviewer #2: Authors have incorporated all suggestions raised for Results, Figures, and Method Sections, which have improved flow and clarity of the manuscript. In addition, author have validated another IKBKG protein which is important for NF-B pathway and discussed their results nicely.

Minor comments: Although, authors have added statistical analysis information in the figures as well as legends, but it appears vague. Is their any rational to use two different statistics tests for example?

• Figure2: Error bars represent S.D. All 420 statistics were calculated using a one-sample t and Wilcoxon test. (*) indicates 421 statistical significance (p<0.05).

• Figure3: All quantified data are presented as mean � S.D. 421 All statistics were calculated using a two-way ANOVA with a Tukey’s multiple 422 comparisons test. (*) indicates statistical significance (p<0.05)

Author should add detail statistical analysis methods information for each panel & how they were compared in ‘Statistics’ section.

Reviewer #3: The authors have responded to the previous reviews and the manuscript is now acceptable. I have no other comments.

7. PLOS authors have the option to publish the peer review history of their article (what does this mean?). If published, this will include your full peer review and any attached files.

Reviewer #1: **Yes: **KARTHIKEYAN THIRUGNANAM

Reviewer #2: No

Reviewer #3: No

---

## [Author Response · Author response to Decision Letter 1]

2 Jun 2021

Please see the attached formatted "Response to Reviewers" letter, which contains the following content:

May 30, 2021

Ramani Ramchandran

Academic Editor

PLOS ONE

RE: Resubmission of PONE-D-20-35540R1

Dear Dr. Ramchandran:

Thank you for the recent reviews of our revised manuscript “Genomic locus proteomic screening identifies the NF-�B signaling pathway components NF�B1 and IKBKG as transcriptional regulators of Ripk3 in endothelial cells.” Reviewer #2 requested a more thorough description of our statistical methods; we appreciated this suggestion and have included responses below:

Reviewer #2:

Minor comments: Although, authors have added statistical analysis information in the figures as well as legends, but it appears vague. Is there any rational to use two different statistics tests for example?

Figure 2: Error bars represent S.D. All statistics were calculated using a one-sample t and Wilcoxon test. (*) indicates statistical significance (p<0.05).

 We chose to use a one-sample t and Wilcoxon test to determine if the mean of ChIP enrichment (for NF�B1 or for IKBKG) is statistically different from the normalized negative IgG control. This test was chosen because we were comparing one variable: each mean enrichment value (independently) against the calculated IgG control enrichment levels.

Figure 3: All quantified data are presented as mean S.D. all statistics were calculated using a two-way ANOVA with a Tukey’s multiple comparisons test. (*) indicates statistical significance (p<0.05).

We chose to use a two-way ANOVA with a Tukey post hoc test because we were comparing two factors simultaneously: siRNAs (nonspecific or gene-specific) and treatment (vehicle or TNF� or Z-VAD-FMK or NSA).

Author should add detail statistical analysis methods information for each panel and how they were compared in “Statistics” section.

We have updated the “Statistics” section within the “Materials and methods” to say the following (new lines 259-268): 

Results are presented as mean � S.D. of n independent experiments (n is reported in the figure legends). Data normality was determined using the D'Agostino-Pearson omnibus test. Two-way ANOVA with a Tukey post hoc test was used to determine changes in HUVECs knocked down with nonspecific siRNA or gene of interest-specific siRNA after treatment with vehicle or TNF� or Z-VAD-FMK or NSA (Figs 2A, 2C, 2D, 3B). One-sample t and Wilcoxon test was used to determine fold-change of ChIP enrichment compared to the normalized negative IgG control (Figs 4A and 4B). Significance was determined by a P-value of 0.05 or less. All statistical analyses were achieved using GraphPad Prism8 software.

Thank you for this opportunity to improve the statistical descriptions associated with this revised manuscript.

Sincerely,

Courtney Griffin, Ph.D.

---

## [Editor Report · Decision Letter 2]

8 Jun 2021

Genomic locus proteomic screening identifies the NF-kB signaling pathway components NFkB1 and IKBKG as transcriptional regulators of Ripk3 in endothelial cells

PONE-D-20-35540R2

Dear Dr. Griffin,

We’re pleased to inform you that your manuscript has been judged scientifically suitable for publication and will be formally accepted for publication once it meets all outstanding technical requirements. *In the proofs, please include the details of the rationale for the statistical tests as you provided in the response to reviewer 2 comments. This will help in understanding the choice of statistical test. *

Kind regards,

Ramani Ramchandran

Academic Editor

PLOS ONE
---

## [Editor Report · Acceptance letter]

10 Jun 2021

PONE-D-20-35540R2 

Genomic locus proteomic screening identifies the NF-kB signaling pathway components NFκB1 and IKBKG as transcriptional regulators of *Ripk3* in endothelial cells 

Dear Dr. Griffin:

I'm pleased to inform you that your manuscript has been deemed suitable for publication in PLOS ONE. Congratulations! Your manuscript is now with our production department. 

Kind regards, 

on behalf of

Dr. Ramani Ramchandran 

Academic Editor

PLOS ONE